# Engineering of ATP synthase for enhancement of proton-to-ATP ratio

Hiroshi Ueno ®[1] ✉, Kiyoto Yasuda[1], Norie Hamaguchi-Suzuki[2,3], Riku Marui[1], Naruhiko Adachi ®[4,5], Toshiya Senda ®[4], Takeshi Murata ®[2] & Hiroyuki Noji ®[1] ✉

$F_oF_1$-ATP synthase ($F_oF_1$) interconverts the energy of the proton motive force (*pmf*) and that of ATP through the mechanical rotation. The $H^+$/ATP ratio, one of the most crucial parameters in bioenergetics, varies among species due to differences in the number of $H^+$-binding c-subunits, resulting in $H^+$/ATP ratios ranging from 2.7 to 5. In this study, we seek to significantly enhance the $H^+$/ATP ratio by employing an alternative approach that differs from that of nature. We engineer $F_oF_1$ to form multiple peripheral stalks, each bound to a proton-conducting a-subunit. The engineered $F_oF_1$ exhibits an $H^+$/ATP ratio of 5.8, surpassing the highest ratios found in naturally occurring $F_oF_1$s, enabling ATP synthesis under low *pmf* conditions where wild-type enzymes cannot synthesize ATP. Structural analysis reveals that the engineered $F_oF_1$ forms up to three peripheral stalks and a-subunits. This study not only provides valuable insights into the $H^+$-transport mechanism of $F_oF_1$ but also opens up possibilities for engineering the foundation of cellular bioenergetics.

$F_oF_1$-ATP synthase ($F_oF_1$) is a ubiquitous enzyme found in the membranes of mitochondria, chloroplasts and bacteria. It synthesizes ATP from ADP and inorganic phosphate (Pi) coupled with proton translocation across membranes along the proton motive force (*pmf*)[1–3]. $F_oF_1$ is a unique molecular motor complex composed of two rotary molecular motors: $F_1$ and $F_o$[1,2]. Bacterial $F_oF_1$ exhibits the simplest subunit composition of $a_1b_2c_x$ (x varies among species) for $F_o$ and $\alpha_3\beta_3\gamma_1\delta_1\epsilon_1$ for $F_1$[4–6] (Fig. 1a). $F_o$ is a membrane-embedded molecular motor driven by *pmf*. When protons are translocated through the proton pathway in $F_o$ along *pmf*, the multimeric rotor ring composed of c-subunits (termed c-ring) rotates against stator $ab_2$ (Fig. 1a, b). $F_1$ is the catalytic core domain of $F_oF_1$ for ATP synthesis and hydrolysis[7]. When isolated from $F_o$, $F_1$ acts as an ATP-driven molecular motor that rotates the γε rotor complex against the catalytic $\alpha_3\beta_3$ stator ring coupled with ATP hydrolysis[8]. These two motors are coupled via two stalks: the peripheral stalk, composed of the $b_2$ dimer stalk and the δ subunit. The central rotor stalk is composed of a γε complex and a c-ring. Under ATP synthesis conditions where *pmf* is sufficient and the rotational torque

of $F_o$ exceeds that of $F_1$, $F_o$ rotates the γε complex in $F_1$ in the reverse direction of ATP hydrolysis, driving the ATP synthesis reaction on the $\alpha_3\beta_3$ ring[9,10]. Conversely, when the torque of $F_1$ exceeds that of $F_o$, $F_1$ rotates the c-ring in $F_o$, forcing $F_o$ to pump protons in the reverse direction, thereby generating *pmf*. Thus, $F_oF_1$ interconverts the *pmf* and chemical potential of ATP hydrolysis through mechanical rotation.

Considering the Gibbs free energy of this coupling reaction ($\Delta G'$),

$$ADP + Pi + nH_{in}^+ \rightleftharpoons ATP + H_2O + nH_{out}^+ \tag{1}$$

$\Delta G'$ is givens:

$$\Delta G' = \Delta G'_{ATP} - nF \cdot pmf \tag{2}$$

where $\Delta G'_{ATP}$ is the Gibbs free energy of ATP synthesis, $F$ is Faraday's constant, $n$ is the $H^+$/ATP ratio, which is defined as the number of protons translocated through $F_o$ coupled with a single turnover of ATP synthesis on $F_1$. Hence, the following conditions must be satisfied to

[1]Department of Applied Chemistry, Graduate School of Engineering, The University of Tokyo, Tokyo, Japan. [2]Department of Chemistry, Graduate School of Science, Chiba University, Chiba, Japan. [3]Department of Pharmacology, Graduate School of Medicine, Chiba University, Chiba, Japan. [4]Structural Biology Research Center, Institute of Materials Structure Science, High Energy Accelerator Research Organization (KEK), Ibaraki, Japan. [5]Life Science Center for Survival Dynamics, Tsukuba Advanced Research Alliance (TARA), University of Tsukuba, Ibaraki, Japan. ✉e-mail: hueno@g.ecc.u-tokyo.ac.jp; hnoji@g.ecc.u-tokyo.ac.jp

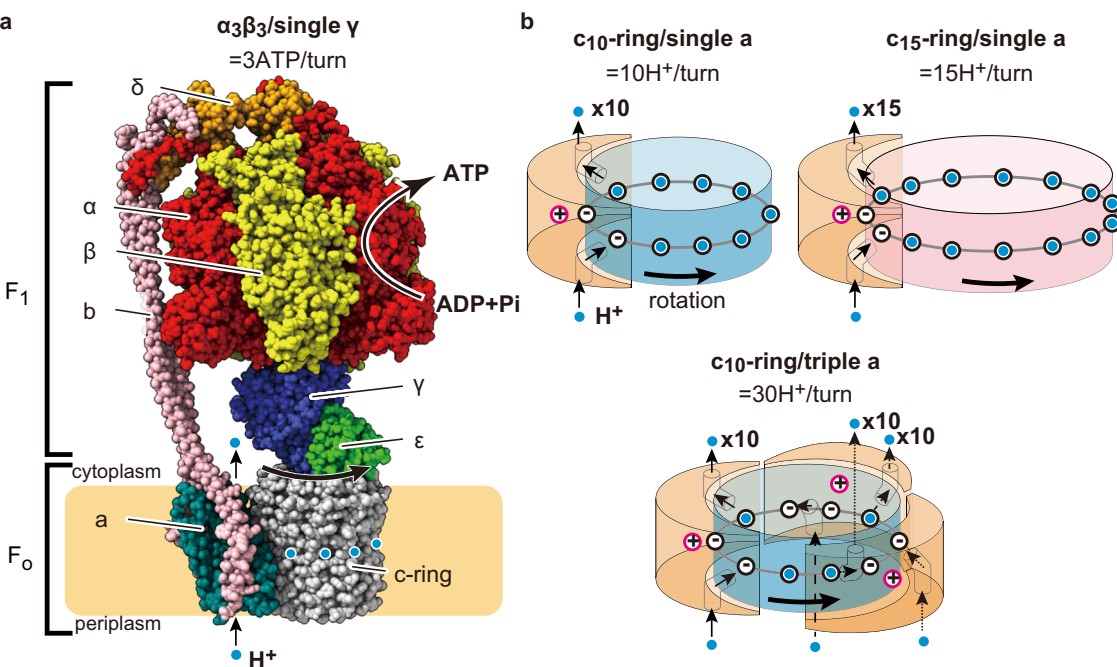

**Fig. 1 | Structure of $F_oF_1$-ATP synthase and rotation mechanism of $F_o$ with a different number of the c-/a-subunits. a** Bacterial ATP synthase (from thermophilic *Bacillus* PS3) consists of $F_1$ ($\alpha_3\beta_3\gamma\delta\epsilon$) and $F_o$ ($ab_2c_{10}$) motors. As $F_1$ has three catalytic sites, three ATP molecules are synthesized per turn of the rotor subunits ($\gamma\epsilon c_{10}$) against the stator subunits ($\alpha_3\beta_3\delta ab_2$) during ATP synthesis. **b** Models of proton translocation through $F_o$ coupled with the rotation of the c-ring. The highly conserved arginine residues of the a-subunit and glutamate (or aspartate) residues

of c-subunits are depicted with pink and black open circles, respectively. Protons are depicted as light blue circles. In the models of $c_{10}$-ring/single a-subunit and $c_{15}$-ring/single a-subunit, 10 and 15 protons, equal to the number of the c-subunits, are transferred in one turn, respectively. Conversely, in the $c_{10}$-ring and triple a-subunits model, a total of 30 protons, equal to the number of the c-subunits multiplied by the number of the a-subunits, are transferred in one turn.

drive ATP synthesis:

$$\Delta G'_{ATP} < nF \cdot pmf \qquad (3)$$

Thus, the H+/ATP ratio is the critical factor in determining the lower limit of *pmf* required for ATP synthesis, given that $\Delta G'_{ATP}$ does not largely vary among species. The H+/ATP ratio is principally defined by the ratio of the reaction stoichiometry of $F_o$ per turn of the rotor complex to that of $F_1$, and the ratio of H+/turn of $F_o$ to ATP/turn of $F_1$. All the $F_1$s studied so far, without exception, have three catalytic β subunits and couple three reactions of ATP hydrolysis/synthesis per turn, defining the ATP/turn ratio as 3[10]. The reaction stoichiometry of $F_o$, H+/turn varies among species due to differences in the number of c-subunits in the c-ring. According to the half-channel model, supported by recent structural studies, the H+ pathway in $F_o$ is formed by the c-ring and the a-subunit, which has two half-channels exposed to the periplasmic or cytoplasmic side of the membrane[11–13]. During ATP synthesis, H+ from the periplasmic solution enters the half-channel exposed to the periplasmic space and is transferred to one of the c-subunits in the c-ring. Following one revolution of the c-ring, H+ is released into the cytoplasmic solution through the opposite half-channel (Fig. 1b, Supplementary Fig. 1). Thus, the half-channel model assumes that the stoichiometry of H+/turn is determined by the number of c-subunits.

The number of c-subunits in the c-ring in F-type ATP synthases ranges from 8 to 15, depending on the species[14–16]. When assuming perfect energy coupling between $F_1$ and $F_o$, the H+/ATP ratio should vary between 2.7 and 5.0 among the species. Various groups have attempted to experimentally determine the H+/ATP ratio from the biochemical measurements of the thermodynamic equilibrium point where the *pmf* and $\Delta G'_{ATP}$ are balanced. The *Bacillus* PS3 $F_oF_1$, with a $c_{10}$-

ring, has been reported to show good agreement with the structurally expected H+/ATP ratio of 3.3[17]. $F_oF_1$s from *E. coli* and yeast mitochondria, both of which also have the $c_{10}$-ring, exhibit slightly different H+/ATP ratios: $4.0 \pm 0.3$[18] and $2.9 \pm 0.2$[19], respectively. For spinach chloroplast $F_oF_1$ with the $c_{14}$-ring, two independent studies reported smaller values for the H+/ATP ratio: $4.0 \pm 0.2$[18] and $3.9 \pm 0.3$[19], which are lower than the expected value of 4.7. Thus, the experimentally determined H+/ATP ratios are close to, but not always identical to, the structurally expected values, varying within a narrow range of 3 to 4.

The H+/ATP ratio of $F_oF_1$ is one of the most critical parameters in the bioenergetic system of cells, which defines the energy cost of ATP synthesis and the threshold *pmf* required for ATP synthesis (see Eq. 3)[14,17]. Since $\Delta G'_{ATP}$ does not largely differ across organisms, $F_oF_1$ with a higher H+/ATP ratio can synthesize ATP even at a lower *pmf*. In fact, alkaliphilic bacteria living in highly alkaline environments and photosynthetic organisms that grow under light-limiting conditions have a c-ring with a large number of c-subunits[20–22]. This is thought to be an evolutionary adaptation that allows stable ATP synthesis under low and/or unstable *pmf* conditions[22,23]. Thus, organisms may have optimized the H+/ATP ratio through evolution by tuning the stoichiometry of the c-ring to meet their energetic requirements.

Conversely, when reconsidering the half-channel mechanism, we can assume that the stoichiometry of H+/turn in $F_o$ is determined not only by the number of c-subunits but also by the number of a-subunits (Fig. 1b, bottom). In particular, the $F_o$ structures solved so far show that a large portion of the c-ring is exposed to the lipid bilayer, suggesting the possibility of accommodating one or two additional a-subunits, although $F_oF_1$ with multiple a-subunits has not yet been identified.

Here, we explore the possibility of doubling or tripling the H+/ATP ratio of $F_oF_1$ by increasing the number of a-subunits in $F_o$, rather than increasing the stoichiometry of the c-ring as observed in nature.

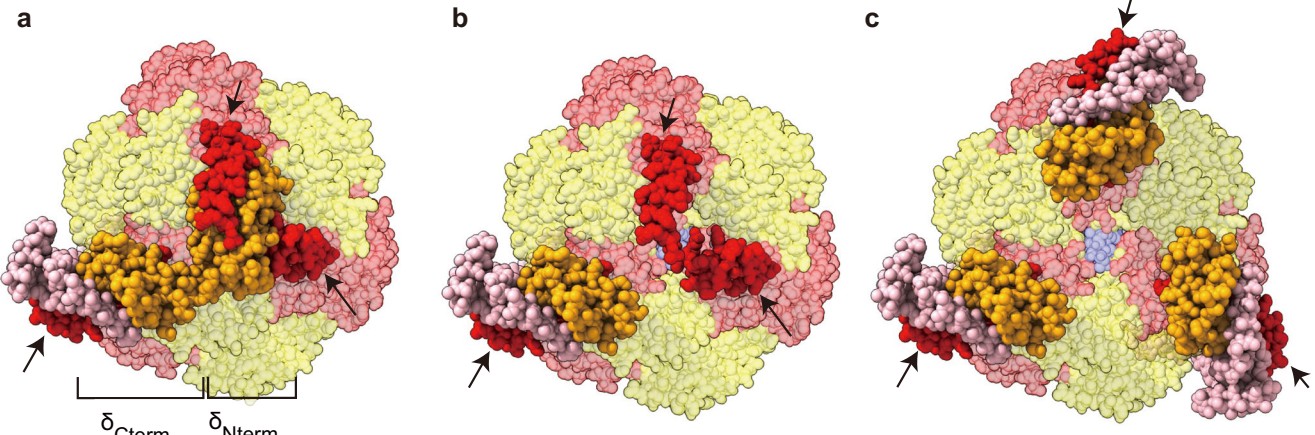

**Fig. 2 | Design strategy to multiply peripheral stalks.** Top views of *Bacillus* PS3 $F_oF_1$ from the cytoplasm (PDB ID: 6N2Y). The α (red), β (yellow), γ (blue), δ subunits (orange), and the $b_2$ stalk (pink) are shown. The non-transparent red parts indicated by the arrows represent the N-terminal region (residues 2–30) of the α subunit. **a** Asymmetric interactions between the δ and α subunits. The single δ subunit interacts with the three α subunits, occupying the central concavity of the $α_3β_3$ subcomplex. **b** Model structure after deletion of the N-terminal domain (residues 2–104) of the δ subunit. The central concavity is exposed, and two of three α subunits are unoccupied. **c** Model structure after each of the N-terminus of the three α subunits is fused to the C-terminal domain (residues 105–178) of the δ subunit.

## Results

### Design for multiple peripheral stalks

So far, all ATP synthases have a single copy of the a-subunit per $F_oF_1$ complex. Considering that the a-subunit is tightly bound to the membrane portion of the peripheral stalk, the key factor for multiplying the a-subunits should be the structural mechanism that limits the number of peripheral stalks. The peripheral stalk of the $b_2$ dimer extends from the membrane to the upper surface of the $α_3β_3$ subcomplex of $F_1$, binding to the δ subunit. The δ subunit binds to the top of the $α_3β_3$ subcomplex, associating with the N-terminal regions of the three α subunits. One α subunit interacts with the C-terminal region of the δ subunit, while the other two interact with the N-terminal region of the δ subunit, as indicated by the arrows in Fig. 2a. Since the N-terminal domain of the δ subunit is located in the central concavity of the $α_3β_3$ ring, occupying the pseudo-threefold symmetry axis of the $α_3β_3$ ring, it is reasonable to assume that the N-terminal domain of the δ subunit disrupts the pseudo-threefold symmetry, limiting the stoichiometry of the peripheral stalk to one (Fig. 2a). We hypothesized that by removing the N-terminal domain of the δ subunit (Fig. 2b), it becomes possible to accommodate a truncated δ subunit on each α subunit (Fig. 2c). A possible concern regarding the truncation is that the truncated δ subunit ($δ_{ΔN}$) does not form a stable complex with the $α_3β_3$ ring. Therefore, we designed the $δ_{ΔN}$-α fusion construct of *Bacillus* PS3 $F_oF_1$, where the C-terminus of the δ subunit was genetically fused to the N-terminus of the α subunit. In addition, the inhibitory C-terminal domain of the ε subunit was removed to enhance enzymatic activity[17,24]. In this study, *Bacillus* PS3 $F_oF_1$-$ε_{ΔC}$ was used as a wild-type $F_oF_1$ for comparison.

### SDS-PAGE analysis of subunit stoichiometry

The $δ_{ΔN}$-α fused $F_oF_1$ was purified following a previously reported procedure for the wild-type *Bacillus* PS3 $F_oF_1$[17]. To estimate the subunit stoichiometry, the $δ_{ΔN}$-α fused $F_oF_1$ was analyzed by SDS-PAGE with the wild-type $F_oF_1$ for comparison (Fig. 3a, b). The $δ_{ΔN}$-α fused $F_oF_1$ lacked δ and α, and the $δ_{ΔN}$-α fusion appeared above the band position for the α subunit (Fig. 3a). The $δ_{ΔN}$-α fused $F_oF_1$ retained the complete set of subunits. Then, we estimated the subunit stoichiometry of the a- and b-subunits in the $δ_{ΔN}$-α fused $F_oF_1$ by using the γ subunit as the internal reference in comparison with the wild-type (Fig. 3b, Supplementary Fig. 2). In the wild-type, the b-subunit exhibited a band intensity

comparable to that of the γ subunit, whereas in the $δ_{ΔN}$-α fused $F_oF_1$, the b-subunit showed higher signals relative to the γ subunit. Similarly, the $δ_{ΔN}$-α fused $F_oF_1$ exhibited a higher band intensity for the a-subunit compared to the wild type, indicating that the $δ_{ΔN}$-α fused $F_oF_1$ increases the stoichiometry of the a- and b-subunits. For a more quantitative estimation, we plotted calibration lines for the wild-type and mutant subunits and standardized the lines using the γ subunit calibration lines as the internal control (Fig. 3b). We then determined the stoichiometries of the a- and b-subunits in the $δ_{ΔN}$-α fused $F_oF_1$ by comparing them to the wild-type. The estimated stoichiometries of the a- and b-subunits were 2.2 and 1.8 times higher than those of the wild-type $F_oF_1$, respectively (see Fig. 3b legend). Thus, it was confirmed that the $δ_{ΔN}$-α fused $F_oF_1$ has multiple, two on average, peripheral stalks and a-subunits.

### Functional analysis of the H⁺/ATP ratio

We attempted to determine the H⁺/ATP ratio of the $δ_{ΔN}$-α fused $F_oF_1$ through biochemical measurements of the thermodynamic equilibrium between *pmf* and $ΔG'_{ATP}$, as previously reported[17]. Firstly, we prepared the $F_oF_1$-reconstituted proteoliposomes (PLs) and incubated them in an acidic buffer. The PLs were injected into the base assay medium to initiate ATP synthesis. The ATP synthesis/hydrolysis activity was monitored with the luciferin/luciferase assay system under various *pmf* conditions, with a given reaction quotient, $Q$ ($=$ [ATP]/([ADP] · [Pi])). Figure 4a shows the time courses of the assay, in which the initial rate was determined. Figure 4b shows the reaction rates plotted against the *pmf* when $Q = 2.5$. The $δ_{ΔN}$-α fused $F_oF_1$ was shown to catalyze the ATP synthesis reaction even at low *pmf*, at which wild-type $F_oF_1$ is unable to synthesize ATP. For a more quantitative analysis, the data points were fitted with an exponential function to determine the equilibrium *pmf* ($pmf_{eq}$), where the torques of $F_1$ and $F_o$ are balanced and the net reaction rate is zero. At the condition of Fig. 4b where $Q = 2.5$, $pmf_{eq}$ was determined to be 68 mV for the $δ_{ΔN}$-α fused $F_oF_1$ and 133 mV for the wild-type. Thus, the minimum *pmf* for ATP synthesis was halved for the $δ_{ΔN}$-α fused $F_oF_1$, suggesting that the functional H⁺/ATP ratio of the $δ_{ΔN}$-α fused $F_oF_1$ is also doubled, in agreement with SDS-PAGE analysis. For further confirmation, we determined the $pmf_{eq}$ at various $Q$ values (Supplementary Fig. 3). Under all conditions, the $pmf_{eq}$ of the $δ_{ΔN}$-α fused $F_oF_1$ was nearly half that of the wild-type.

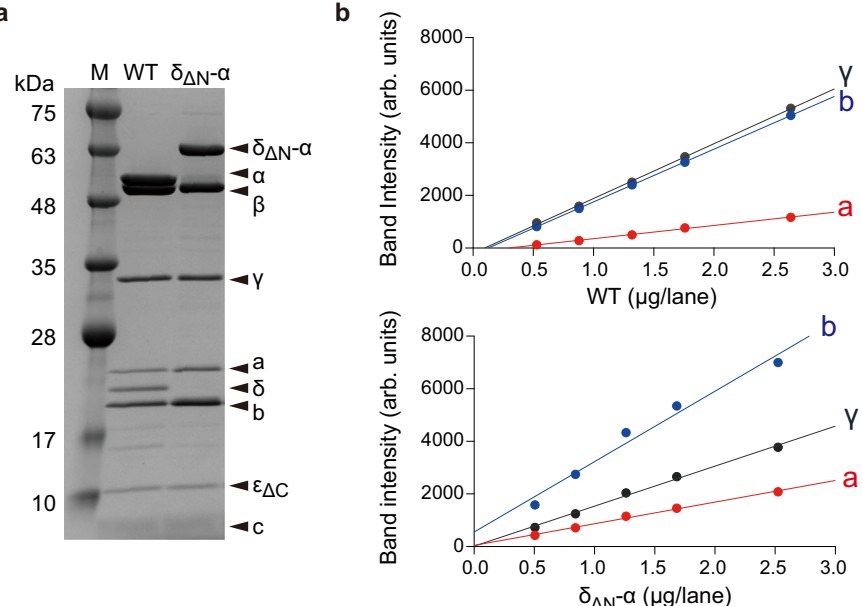

**Fig. 3 | Subunit stoichiometry of $\delta_{\Delta N}$-$\alpha$ fused F$_o$F$_1$. a** SDS-PAGE analysis of the purified wild-type (WT) *Bacillus* PS3 F$_o$F$_1$ and the $\delta_{\Delta N}$-$\alpha$ fused F$_o$F$_1$. Each sample was derived from a single purification batch. 3 μg of F$_o$F$_1$ was loaded in each lane. The molecular masses of the $\delta_{\Delta N}$-$\alpha$, $\alpha$, $\beta$, $\gamma$, a, $\delta$, b, $\varepsilon_{\Delta C}$, and c-subunits are 63, 55, 53, 32, 26, 20, 19, 9, and 7 kDa, respectively. The experiment was independently repeated three times with similar results. **b** The band intensity vs total protein amount. A single series of diluted F$_o$F$_1$ from the same purification batch was loaded into a gel and subjected to SDS-PAGE analysis. The band intensity of each subunit was plotted against the total amount of F$_o$F$_1$ loaded for SDS-PAGE analysis. The plots were fitted with a linear function. The slopes for $\gamma$ (black), a- (red), and b-subunits (blue) were determined to be 2082, 502, and 2004 (arb. units/μg) for the WT F$_o$F$_1$, and 1521, 821, and 2675 (arb. units/μg) for the $\delta_{\Delta N}$-$\alpha$ fused F$_o$F$_1$, respectively. By normalizing the slopes of the a- and b-subunits to the slope of the $\gamma$ subunit of each F$_o$F$_1$, the stoichiometries of the a- and b-subunits of the $\delta_{\Delta N}$-$\alpha$ fused F$_o$F$_1$ were estimated to be 2.2 ( = (821/1521)/(502/2082)) and 1.8 ( = (2675/1521)/(2004/2082)) times higher than those of the wild-type F$_o$F$_1$, respectively.

For the comprehensive analysis of the functional H$^+$/ATP ratio, Eq. (2) was transformed as below,

$$\Delta G' = \Delta G'_{ATP} - nF \cdot pmf = \Delta G^{0'}_{ATP} + 2.3RT \cdot \log Q - nF \cdot pmf_{eq} = 0 \quad (4)$$

$$2.3RT \cdot \log Q = -\Delta G^{0'}_{ATP} + nF \cdot pmf_{eq} \quad (5)$$

where $\Delta G^{0'}_{ATP}$ is the Gibbs free energy of ATP synthesis under the biochemical standard state, $R$ and $T$ are the gas constant and absolute temperature, respectively. Here, $pmf_{eq}$ values were experimentally determined under defined $Q$ conditions. The other values are constant. Therefore, when $2.3RT \cdot \log Q$ is plotted against $F \cdot pmf_{eq}$, $n$ (= the H$^+$/ATP ratio) is determined as the slope of the data points. In addition, $G^0_{ATP}$ is determined as the interception of the y axis. As shown in Fig. 4c, the $\delta_{\Delta N}$-$\alpha$ fused F$_o$F$_1$ exhibited a significantly steeper slope compared to the wild-type F$_o$F$_1$. From the linear fitting, the H$^+$/ATP ratio was determined to be 5.8 ± 0.4 and 3.0 ± 0.2 (fitted value ± SE of the fit) for the $\delta_{\Delta N}$-$\alpha$ fused F$_o$F$_1$ and wild-type, respectively. Although the H$^+$/ATP ratio of the wild-type is slightly lower than the structurally expected value of 3.3 and the reported value (3.3 ± 0.1)[17], $\delta_{\Delta N}$-$\alpha$ fused F$_o$F$_1$ was shown to double the H$^+$/ATP ratio, in close agreement with the subunit stoichiometry analysis from SDS-PAGE. This agreement suggests that the $\delta_{\Delta N}$-$\alpha$ fused F$_o$F$_1$ has two functional a-subunits in the ensemble average. From the interception of the y axis, the $\Delta G^0_{ATP}$ values are determined to be 38 ± 3 kJ mol$^{-1}$ (fitted value ± SE of the fit) for both. This value shows a fine agreement with the reported values for *Bacillus* PS3 F$_o$F$_1$ (39 ± 1 kJ mol$^{-1}$)[17], *E. coli* (38 ± 3 kJ mol$^{-1}$)[18], yeast (36 ± 3 kJ mol$^{-1}$)[19], and chloroplasts (38 ± 3 and 37 ± 3 kJ mol$^{-1}$)[18,19], supporting the validity of the experiment.

### Cryo-EM structural analysis

We determined the structure of the $\delta_{\Delta N}$-$\alpha$ fused F$_o$F$_1$ by single-particle cryo-EM analysis. The purified $\delta_{\Delta N}$-$\alpha$ fused F$_o$F$_1$ in detergent was applied to EM grids, frozen in liquid ethane, and imaged with 300 kV cryo-EM followed by single-particle analysis using cryoSPARC. The cryo-EM map was obtained by ab initio 3D reconstruction and classification, followed by refinement with C1 symmetry. Since the rotor complex in F$_o$F is oriented at one of the three catalytic dwell angles relative to the stator ring and peripheral stalk, alignment against the central core complex, including the rotor complex, revealed three distinct positions of the peripheral stalk, each separated by 120°, as observed in previous reports[25,26]. Map structure classification was performed by masking the peripheral-stalk positions, confirming the presence or absence of the peripheral stalk at the masked position (Supplementary Fig. 4). This classification was conducted for each stalk position: Stalks 1, 2, and 3. Thus, the map structure was classified into eight sub-classes. The overall resolution was 2.5–3.2 Å (Supplementary Fig. 4). While the wild-type F$_o$F$_1$ contains only a single peripheral stalk, this structural classification confirmed that some fractions of the particles contained multiple peripheral stalks (Fig. 5 and Supplementary Fig. 4). A significant fraction of the particles, however, had either no peripheral stalk or only a single one. The percentages of F$_o$F$_1$ structures with 0, 1, 2, and 3 peripheral stalks, as determined from the 3D classification, were 15, 51, 26, and 8%, respectively. Because of the lower resolution of the F$_o$ region, focused refinement with F$_o$ was conducted by masking the F$_o$ region. The refined structure achieved resolutions of 3.5–6.6 Å, confirming that the peripheral stalk always accompanies F$_o$ a-subunits (Supplementary Fig. 4). Thus, the percentage of F$_o$F$_1$ with 0, 1, 2, and 3 a-subunits should correspond to that for peripheral stalks. The fractions of F$_o$F$_1$ with multiple a-subunits were small, and the ensemble average of peripheral stalk was only 1.26 per molecule, which is evidently lower than the expected value of ~2 per molecule based on the subunit stoichiometry analysis and the functional analysis of H$^+$/ATP ratio. Although the exact reason for this discrepancy is unclear, it is highly likely that the peripheral stalk and the a-subunit dissociated due to the meniscus force

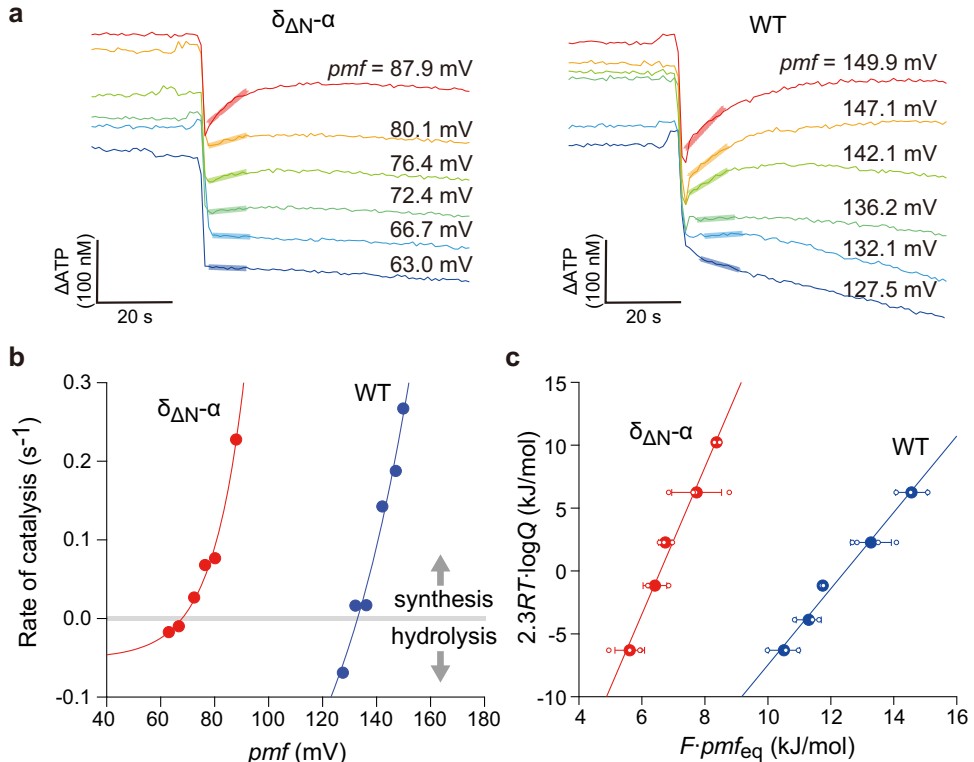

**Fig. 4 | Functional analysis for the determination of the H⁺/ATP ratio. a** Time courses of the ATP synthesis/hydrolysis activity of the reconstituted proteoliposomes (PLs) at different *pmf*. ATP synthesis reaction was measured using the luciferin/luciferase system. The reaction quotient, *Q* was 2.5; [ATP] = 500 nM, [ADP] = 20 μM, [Pi] = 10 mM. The rate of catalysis was determined from the initial slopes (bold lines). **b** The ATP synthesis/hydrolysis rates determined from (**a**) were plotted against *pmf*. The data points were fitted with an exponential function for the determination of the equilibrium *pmf*, $pmf_{eq}$, as the interception of the *x* axis.

**c** Determination of the H⁺/ATP ratio. The mean (filled circles) and the SD of $F \cdot pmf_{eq}$ values were determined from 3 to 4 independent biological replicates (open circles) at each *Q* condition using different batches of purified enzymes, and $2.3RT \cdot \log Q$ values were plotted against the corresponding $F \cdot pmf_{eq}$ values according to Eq. (5). Sample sizes (*n*) from left to right are: 3, 3, 3, 4, 3 (WT) and 4, 3, 3, 4, 3 ($\delta_{\Delta N}$-α). Each line represents a linear regression fit to the dataset obtained from each $F_oF_1$.

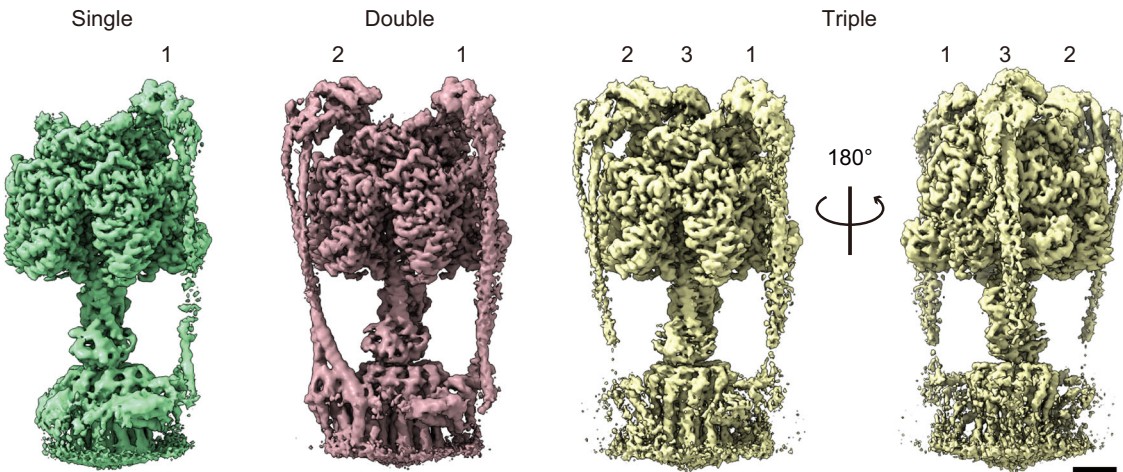

**Fig. 5 | Cryo-EM maps of $F_oF_1$ with multiple peripheral stalks.** The composite cryo-EM maps of $F_oF_1$ with single, double, and triple peripheral stalks. The number represents the position of each peripheral stalk. Scale bar, 25 Å.

and/or the interaction with the air/water interface during cryo-EM grid preparation[27].

## Interaction between the peripheral stalk and the $\delta_{\Delta N}$-α fusion

Among the eight sub-classes, three structures contained a single copy of the peripheral stalk, each located at position Stalk 1, Stalk 2, or Stalk 3. These structures fit well with the reported three rotational state

structure of the wild-type *Bacillus* PS3 $F_oF_1$[25], respectively (Supplementary Fig. 5), indicating the structural integrity of the binding of the $b_2$ dimer to the $F_1$ part via $\delta_{\Delta N}$-α fusion. Slight differences were observed at the top of the $F_1$ headpiece and on the side of the central stalk because the $\delta_{\Delta N}$-α fused $F_oF_1$ lacks the N-terminal region of the δ subunit and the C-terminal helix of the ε subunit. The maps of the double- and triple-stalk $F_oF_1$ were compared with the corresponding

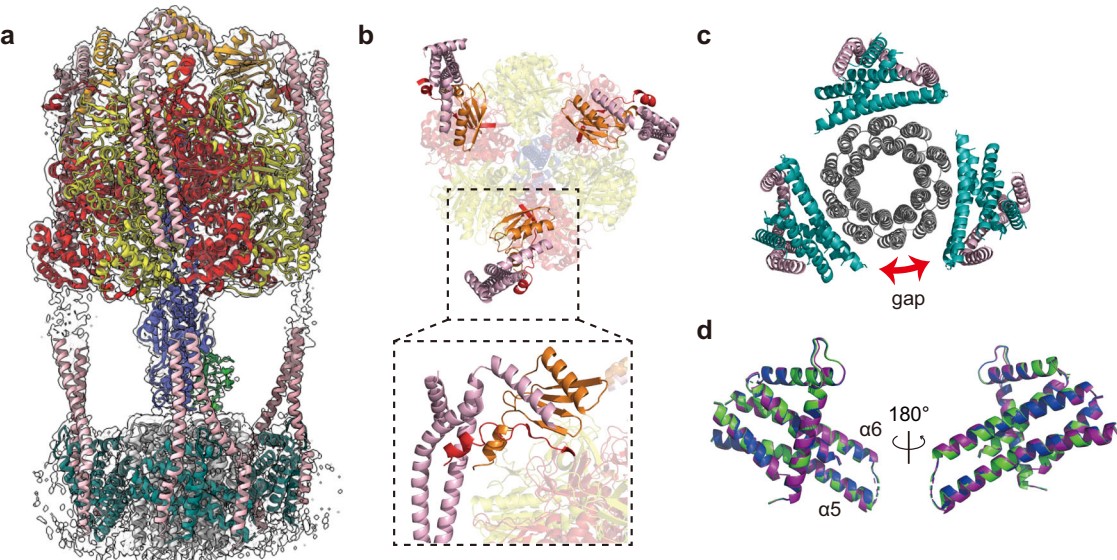

**Fig. 6 | Atomic models of $F_oF_1$ with triple peripheral stalks. a** Composite map and atomic models for the $F_1$ and $F_o$ regions of the triple-stalk $F_oF_1$. The middle regions of the b-subunits could not be modeled due to the lack of clear density. **b** The top view of the structure. The close-up view shows the side view of the interaction between $\delta_{\Delta N}$-α fusion and the $b_2$ dimer. **c** The view from the bottom of the structure shown in (**a**). **d** The superposition of the three a-subunits (green, purple, and blue) in the triple-stalk $F_o$.

maps of single-stalk $F_oF_1$, respectively (Supplementary Fig. 6). These maps were well fitted, indicating that multiple stalks have no significant structural constraints on the whole structure of $F_oF_1$.

The atomic models for triple-stalk $F_oF_1$ are shown in Fig. 6. The structure of the $\delta_{\Delta N}$-α fusion region was well resolved, providing atomic details of its binding site with the $b_2$ dimer (Fig. 6a, b). As designed, the three binding sites were almost identical, with structures similar to those found in wild type *Bacillus* PS3 $F_oF_1$[25] and other ATP synthases[28] (Supplementary Fig. 7a). These observations reveal that the three $b_2$ dimers are incorporated via the canonical interaction with $\delta_{\Delta N}$-*a* fusion, suggesting the integrity of the peripheral stalks of the $\delta_{\Delta N}$-α fused $F_oF_1$.

### Structure of the a-subunits

The structure and the position of the a-subunits of the triple-stalk $F_oF_1$ were investigated by comparing them with those in the wild-type *Bacillus* PS3 $F_oF_1$[25] (Fig. 6c and Supplementary Fig. 7b). The spatial intervals between the a-subunits are not perfectly symmetric because of the symmetry mismatch between the ring structures of $F_1$ and $F_o$, that is, threefold versus tenfold. Each a-subunit interacted with three neighboring c-subunits, forming an $a_1c_3$ unit. Therefore, a total of nine c-subunits interacted with the a-subunits, leaving the remaining c-subunit at the open position (gap) between two a-subunits (Fig. 6c). As a result, the three a-subunits were not positioned exactly 120° apart from each other. The asymmetric positioning of the a-subunits is in good agreement with that suggested by the three states of the wild-type *Bacillus* PS3 $F_oF_1$[25]. When the a-subunits in the triple-stalk $F_oF_1$ were compared to each other, Cα-RMSDs were 0.3–0.5 Å (Fig. 6d). In addition, the Cα-RMSDs estimated by superimposition of each a-subunit in the triple-stalk $F_oF_1$ with that in the corresponding state of the wild-type $F_oF_1$ were 0.9–1.0 Å. Moreover, superimposing the $a_1c_3$ units in the same manner yielded Cα-RMSDs of 1.0–1.2 Å (Supplementary Fig. 7c). Thus, at the current resolution, the overall structures of the three a-subunits in the triple-stalk $F_oF_1$ were essentially identical to each other and closely resembled that of the wild-type *Bacillus* PS3 $F_oF_1$. This suggests that all a-subunits are functional, which is consistent with the aforementioned biochemical analyses showing the enhanced H+/ATP ratio.

### Structure of $F_1$ part

The structure of the $F_1$ portion of the triple-stalk $F_oF_1$ was investigated by comparing it with previously reported structures. The $F_1$ structure was found to be very similar to that of the *Bacillus* PS3 $F_oF_1$-$\varepsilon_{\Delta C}$ under uni-site catalysis conditions[26], with one β subunit bound to ADP in a closed conformation ($\beta_{TP}^C$) and two β subunits without bound nucleotide adopting open conformations ($\beta_E^O$, $\beta_{DP}^O$) (Supplementary Fig. 8a, b). The purified $\delta_{\Delta N}$-α fused $F_oF_1$ was prepared in nucleotide-free conditions. Therefore, it is highly likely that the bound ADP was endogenous. Notably, at a low-density threshold, weak map density was observed at the outer periphery of $\beta_{TP}^C$, which can be well fitted with the open β conformation of the nucleotide-depleted $F_1$ in the *Bacillus* PS3 $F_oF_1$-$\varepsilon_{\Delta C}$[26] (Supplementary Fig. 8c). This suggests that the cryo-EM map includes two conformations: closed and open. Other subclass structures also exhibited similarly mixed maps.

### Discussion

The present study provides insights into the design principles of ATP synthases using an engineering approach. Firstly, the δ subunit is the factor that defines the number of peripheral stalks. In this study, the N-terminal domain of δ, which breaks the structural pseudo-threefold symmetry by binding to the position on the symmetry axis, was deleted, and the C-terminal domain of δ was genetically fused to the N-terminal of the α subunit. As a result, up to three peripheral stalks were incorporated into the $F_oF_1$ complex. Structural analysis using cryo-EM revealed that the $\delta_{\Delta N}$-α fusion and the $b_2$ dimer adopted the canonical binding structure observed in the wild-type $F_oF_1$, except for the missing N-terminal domain of the δ subunit. This observation clearly shows that the N-terminal domain of the δ subunit determines the number of peripheral stalks per $F_oF_1$, restricting it to one by disrupting the structural symmetry.

Next, the number of a-subunits is determined based on the number of peripheral stalks. While cryo-EM analysis showed that some molecules lost peripheral stalks due to the detachment of the $b_2$ dimer, the observed peripheral stalks always remained bound to the a-subunits, indicating stable binding between the a-subunit and the $b_2$ dimer. Thus, the number of peripheral stalks determines the number of a-subunits. The structure of the triple-stalk $F_oF_1$ clearly showed that the $c_{10}$ ring can accommodate up to three a-subunits, but not more

than four due to spatial constraints. Considering that each a-subunit can interact with three c-subunits, accommodating four or more a-subunits in $F_oF_1$ would require the c-ring consisting of more than 12 c-subunits, along with further engineering on the $F_1$ part, such as introducing a peripheral interaction site at the N-terminus of the β-subunit.

Another important finding of this study is the functional independence of the a-subunit, at least in terms of the coupling stoichiometry of $H^+$ (see below). In the triple-stalk $F_oF_1$, all three a-subunits interacted with the c-ring. In addition, the structural features of the interaction agreed well with those observed in the wild-type $F_oF_1$ structures. This suggests that each of the three a-subunits is functional. SDS-PAGE analysis revealed that the samples used in this study had an average of two peripheral stalks and two a-subunits. Consistent with these results, analysis of the equilibrium *pmf* revealed an $H^+$/ATP ratio that was doubled compared to that of the wild type. These results indicate that the coupling stoichiometry of the $H^+$ ions is proportional to the number of a-subunits. The additivity in $H^+$ stoichiometry means that each a-subunit tightly couples $H^+$ translocation and rotation of the c-ring, regardless of the presence of other a-subunits. This is consistent with the half-channel model, which assumes that the probability of $H^+$ translocation between the a and the c-subunits depends primarily on the relative position of these subunits, which explains well the functional independence of the a-subunits. Based on these considerations, we propose that the number of $H^+$ ions transported coupled with rotation is determined not only by the number of c-subunits constituting the c-ring but also by the number of a-subunits as follows:

$$N_{H^+/turn} = N_c \times N_a \qquad (6)$$

where $N_{H^+/turn}$, $N_c$, and $N_a$ represent the total number of $H^+$ ions per turn, the number of c-subunit in the c-ring, and the number of a-subunit in $F_o$, respectively. However, one might point out the inconsistency between the biochemical results and the structural analysis with cryo-EM. The proportion of molecules with three peripheral stalks was extremely low, clearly lower than the average number suggested by the biochemical results. This is attributable to the dissociation of the peripheral stalks during sample preparation for cryo-electron microscopy. To confirm this point, it is necessary to develop multi-stalk $F_oF_1$ in which peripheral stalks stably bind to $F_1$ and to more accurately analyze the correlation between the number of the a-subunit and the $H^+$ stoichiometry.

As mentioned above, the increased $H^+$/ATP ratio suggests the functional independence of the a-subunits. However, this does not guarantee kinetic independence among the a-subunits. Thus, an arising question is: 'Can $F_o$ with multiple a-subunits rotate the c-ring at the same rate as the wild-type $F_o$?'. In other words, 'Do the a-subunits conduct $H^+$ translocation without mutual interference?' This is a reasonable question considering that, before each 36° rotation step of the c-ring, all a-subunits must complete the $H^+$ translocation with the interacting c-subunits. Therefore, the time constant for each 36° rotation step is expected to be proportional to the number of a-subunits, meaning that the rate constant would decrease inversely.

Our biochemical experiments suggest that ATP hydrolysis-coupled proton pump activity was lower than that of the wild type (Supplementary Table 1). This aligns with the above expectation. However, other scenarios are possible: the engineered $F_oF_1$ may be more susceptible to the *pmf* progressively generated upon proton transport, and the engineered $F_oF_1$ may affect the catalytic activity of $F_1$. Therefore, a quantitative and systematic analysis is necessary to verify this issue.

Regarding the lower activity of the mutant, a naive question could arise: 'Why can the mutant still carry out the ATP synthesis reaction under lower *pmf* conditions despite its lower activity?'. The reason is as follows: the *pmf* required for $F_oF_1$ to synthesize ATP is determined by the $pmf_{eq}$ at which the synthesis and hydrolysis reaction rates are balanced. The present study shows that the engineered $F_oF_1$ with multiple a-subunits doubles the $H^+$/ATP ratio, resulting in a halved $pmf_{eq}$. Therefore, although the engineered ATP synthase has lower catalytic activity, the mutant enzyme can continue the ATP synthesis reaction under low *pmf* conditions where the wild-type enzyme is unable to synthesize ATP.

This study has shown that, in principle, ATP synthase has the capacity to expand its $H^+$/ATP ratio to more than double. To date, experimentally confirmed $H^+$/ATP ratios of $F_oF_1$ have only been ~3–4. Even structural estimates suggest that 5.0 is the maximum for photosynthetic bacteria. Regarding this point, an N-type ATPase—considered a subtype of the F-type—has been reported to have an exceptionally large c-ring composed of 17 c-subunits[29]. Although the function and overall structure of this enzyme remain unknown, its predicted $H^+$/ATP ratio is 5.7. The present study demonstrates that the $H^+$/ATP ratio can be significantly increased by genetically engineering ATP synthase to increase the number of a-subunits, without resorting to such a large c-ring. Such genetic mutations may have arisen over the course of evolution. Indeed, we found that the gene operon of ATP synthase from *Acidaminococcus fermentans* shows the gene fusion of the δ and the α subunits (Supplementary Fig. 9). In addition, the N-terminal domain of δ is missing. These features are well consistent with the $δ_{ΔN}$-α fused $F_oF_1$ we designed. Thus, it is highly likely that *A. fermentans* $F_oF_1$ has a multi-stalk structure and can synthesize ATP under low *pmf* conditions. We also found that other species show similar features (UniProt ID: G4Q3K6, A0A1I2C5T3), suggesting more possibility of a multi-stalk $F_oF_1$ in nature. Thus, the present study suggests the unexpected diversity in the design principles of the $F_oF_1$ ATP synthase, which awaits further experimental verification.

From an engineering standpoint, the findings of the present study may provide future directions for cell engineering. So far, in microbial fermentation, attention has primarily been devoted to developing metabolic pathways and altering metabolic fluxes, while little effort has been made to optimize the intracellular concentrations and ratios of NAD(P)H and ATP, which are fundamental to cellular bioenergetics. When ATP synthesis is driven by oxidative phosphorylation, the amount of ATP produced depends on the extent of NADH oxidation. In such cases, an increase in the $H^+$/ATP ratio translates into a higher NADH/ATP ratio. This could have a significant impact on microbial fermentations. In particular, it could enhance bioproduction in photosynthetic bacteria when ATP is the bottleneck factor[22,30]. To verify this possibility, it will be necessary to investigate how introducing ATP synthase with a modified $H^+$/ATP ratio affects cell growth and product formation. Additionally, such insights are likely to shed light on the physiological role of naturally occurring multi-stalk $F_oF_1$, which may also exist in nature.

## Methods

### Preparation of $F_oF_1$

In this study, *Bacillus* PS3 $F_oF_1$-$ε_{ΔC}$, which has a 10× His-tag at the N-terminus of the β subunit and lacks the inhibitory C-terminal domain of the ε subunit[17,24] was used as the wild-type. The engineered $δ_{ΔN}$-α fusion construct of *Bacillus* PS3 $F_oF_1$-$ε_{ΔC}$ lacks the full-length δ subunit and has a $δ_{ΔN}$-α fused subunit in which the N-terminal domain (residues 2–104) of the δ subunit is deleted and its C-terminus is fused to the N-terminus (without Met) of the α subunit via a short linker. The wild-type and engineered $F_oF_1$s were expressed in *E. coli* DK8 cells, which lack endogenous $F_oF_1$ genes, by incubating in Super broth at 37 °C for 20 h. Cultured cells were suspended in a solution (10 mM HEPES, pH 7.5, 5 mM MgCl₂, and 10% (v/v) glycerol) and disrupted by sonication. After removing the cell debris at $9100 \times g$ for 45 min, membrane fraction was collected by centrifugation for $131{,}500 \times g$ for 1 h at 4 °C. $F_oF_1$ was solubilized from the membrane fraction by adding 0.5% (w/v) LMNG (NG310; Anatrace, USA) and incubating for 30 min at 30 °C. After centrifugation at $162{,}000 \times g$ for 30 min, the solubilized

fraction was applied to a Ni-Sepharose column pre-equilibrated with M buffer (20 mM potassium phosphate buffer and 100 mM KCl, pH 7.5) containing 0.005% LMNG. The column was washed with M buffer containing 20 mM imidazole and 0.005% LMNG, and $F_oF_1$ was eluted with M buffer containing 200 mM imidazole and 0.005% LMNG. The eluted $F_oF_1$ fractions were concentrated before being applied to a Superdex 200 Increase 10/300 column (Cytiva) equilibrated with gel filtration buffer (20 mM HEPES, pH7.5, 100 mM NaCl, and 0.005% LMNG). The peak fractions corresponding to $F_oF_1$ were collected and concentrated to 5–10 mg/mL, frozen with liquid nitrogen, and stored at −80 °C until use. The protein concentrations were determined using a BCA protein assay kit (Pierce) with bovine serum albumin as a standard. The molecular weight of the protein was calculated based on the sequence and subunit stoichiometry. For the $\delta_{\Delta N}$-α fused $F_oF_1$, the molecular weight was calculated assuming an average of two peripheral stalks per $F_oF_1$ molecule.

### Measurement of ATP synthesis/hydrolysis activity of $F_oF_1$

ATP synthesis/hydrolysis activity of $F_oF_1$ was measured using a luciferin-luciferase system at 25 °C, as described previously[17]. $F_oF_1$-reconstituted PLs were prepared as described[17]. Then, 300 μL of the PLs were mixed with 700 μL of acidic buffer containing 50 mM MES or HEPES buffer, 0.143–14.3 mM $NaH_2PO_4$, 6.7 mM KCl, 49 mM NaCl, 4 mM $MgCl_2$, and 600 mM sucrose and NaOH to obtain the desired pH, and then ADP and valinomycin were added to a final concentration of 20–640 μM and 200 nM, respectively. After incubation for 10–24 h at 25 °C for acidification, base assay medium was prepared by mixing 25 μL of the luciferin/luciferase mixture (2× concentration of CLSII solution in ATP bioluminescence assay kit, Roche, and 5 mM luciferin), 800 μL of the base buffer (380 mM HEPES buffer, 0.1125–11.25 mM $NaH_2PO_4$, 5.63 mM KCl, 55 mM NaCl, 4 mM $MgCl_2$, KOH to adjust $K^+$ concentration and NaOH to adjust pH), 50-100 μL of ATP and ADP to obtain the desired concentration, and water to adjust the total volume to 900 μL and incubated for 10 min for equilibrium. Then, the 100 μL of acidified PLs was injected into the base assay medium to initiate the ATP synthesis reaction, and the ATP synthesis/hydrolysis activity was monitored with the luciferin/luciferase assay system using a luminometer (Luminescencer AB2200, ATTO). For calibrating luminescence light intensity to ATP concentration, 10 μL of 10 μM ATP was added. The rate was determined from the initial slope of the linear regression of the time courses. The ΔpH was obtained by subtracting $pH_{in}$ from $pH_{out}$, which was determined by directly measuring the pH using a glass electrode. Transmembrane electrical potential, Δψ, was estimated from the Nernst equation.

### Other assays

SDS-PAGE analysis was performed with 10–20% (w/v) gradient gels. The gels were stained with Coomassie Brilliant Blue (CBB Stain One Super, Nacalai Tesque, Japan) and imaged with a ChemiDoc Imaging System (BIORAD, USA). The band intensity of each subunit was measured using the gel analyzer tool of Fiji (ImageJ 1.54 f) software. ATPase activity measurements of PLs using an ATP regeneration system were performed at 25 °C in the ATPase assay solution (50 mM HEPES-KOH, pH 7.5, 100 mM KCl, 5 mM $MgCl_2$, 2 mM ATP, 1 μg/ml Carbonyl cyanide-p-trifluoromethoxyphenylhydrazone, 2.5 mM phosphoenolpyruvate, 100 μg/mL lactate dehydrogenase, 100 μg/mL pyruvate kinase and 0.2 mM NADH) as described previously[31]. ATP-driven $H^+$-pumping activity was measured by quenching of ACMA (9-amino-6-chloro-2-methoxyacridine) fluorescence at 25 °C in PA4 buffer (10 mM HEPES-KOH, pH 7.5, 100 mM KCl, 5 mM $MgCl_2$) supplemented with 0.3 μg/ml ACMA and 1.0 μg/ml $F_oF_1$-reconstituted PLs[31].

### Cryo-EM grid preparation and data collection

After adding 0.05% lysophosphatidylcholine (1-palmitoyl-2-hydroxy-sn-glycero-3-phosphocholine), 3.0 μL of purified protein (3–6 mg/mL) was loaded onto the glow-discharged Quantifoil R1.2/1.3 grids using a Vitrobot Mark IV (Thermo Fisher Scientific). Grids were blotted for 5 s with a blotting force of 15 under 100% humidity at 18 °C, and flash-frozen in liquid ethane. Data were collected using a 300 kV Titan Krios electron microscope (Thermo Fisher Scientific) with a Falcon 4i direct detector device camera with Selectris-X automated with EPU software. Images were recorded in electron counting mode by recording 50 movie frames with an exposure rate of 1.0 $e^-/Å^2$ per frame. The defocus range was 0.8–2.0 μm, and the original pixel size was 0.75 Å.

### Cryo-EM data processing

All image processing steps were performed using cryoSPARC v4.3.0. Details of the image processing workflow are described in Supplementary Fig. 4. A total of 56,081 micrographs were first motion corrected, and the CTF was estimated by patch CTF estimation. Particles were manually picked, and templates for particle selection were generated from 2D classification. After template picking, the selected 4,316,627 particles were subjected to 2D classification. Then, further selections with ab initio 3D reconstruction, heterogeneous, homogeneous, and non-uniform refinements were performed. After non-uniform refinement, all particles were sequentially subjected to a focused 3D refinement using masks for each of the three peripheral stalks, including the N-terminal region of the α subunit, the C-terminal region of the δ subunit and the hydrophilic region of the b₂-subunits. Each mask was generated from three rotational states of wild-type *Bacillus* PS3 $F_oF_1$ (PDBs 6N2Z, 6N30, and 6N2Y). Eight classes, including one to three peripheral stalks and three rotational states of $F_oF_1$, were identified. The three datasets were collected, merged and refined with non-uniform refinement, resulting in an overall resolution of 2.5–3.2 Å. Further refinement with the $F_o$ mask resulted in a map of the $F_o$ region with an improved resolution of 3.5–6.6 Å. The resolution was estimated using the FSC criterion of 0.143 threshold. The cryo-EM data collection and refinement statistics are shown in Supplementary Tables 2 and 3. The local resolution maps, FSC curves, orientation distribution plots and Model-to-map fits are shown in Supplementary Figs. 10–12.

### Model building

Models were built and refined in COOT v0.9.8.93 and PHENIX v1.20.1-4487. using PDB 6N2Z, 6N30, and 6N2Y as the initial model. Validation statistics are shown in Supplementary Tables 2 and 3. A composite map was generated by combining the $F_1$ region of the triple-stalk $F_oF_1$ map and the $F_o$ map obtained through local refinement with the $F_o$ mask, using UCSF ChimeraX v1.8 for illustration purposes only. Figures were prepared using PyMOL v2.4.0&2.5.2, UCSF Chimera v1.17.3, and UCSF ChimeraX v1.5&1.8. RMSD values for Cα-atoms were calculated using PyMOL 2.4.0 align command without outlier rejection.

### Reporting summary

Further information on research design is available in the Nature Portfolio Reporting Summary linked to this article.

## Data availability

The cryo-EM maps and models generated in this study were deposited to EMDB and PDB under the following accession codes: EMD-61339; EMD-61340; EMD-61341; EMD-61342; EMD-61343; EMD-61344; EMD-61345; EMD-61346; EMD-61347; EMD-61348; EMD-61349; EMD-61350; EMD-61351; EMD-61352; EMD-61353; EMD-61354, and 9JC1; 9JC2. Previously published structures used in this study are also available from PDB under the following accession codes: 6N2Z; 6N30, and 6N2Y. Source data are provided with this paper.

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

## Acknowledgements

We thank all the members of our laboratory for their comments. We also thank N. Soga for technical advice on biochemical experiments. We also thank M. Kawasaki, A. Ikeda, S. Inaba, T. Moriya and the staff at the KEK Structural Biology Research Center for their assistance in collecting and analyzing cryo-EM data. We also thank T. Matsui for technical assistance with the biochemical experiments. This study was supported in part by a Grant-in-Aid for Scientific Research on Innovation Areas (JP21H00388 to H.U.), a Grant-in-Aid for Challenging Research (Exploratory; JP23K18092 to H.U.), a Grant-in-Aid for Scientific Research (B) (JP24K01987 to H.U.), and a Grant-in-Aid for Scientific Research (S) (JP19H05624 to H.N.) from JSPS, and a Research Grant from Human Frontier Science Program (Ref. No: RGP0054/2020 to H.N.), and a JST ASPIRE Program (JPMJAP24B5 to H.N.), and a Research Support Project for Life Science and Drug Discovery (Basis for Supporting Innovative Drug Discovery and Life Science Research (BINDS)) from AMED under Grant Number JP23ama121013 (support number 4318) to T.M. and JP21am0101071 (support number 3071) to T.S.

## Author contributions

H.U., K.Y. and R.M. performed the biochemical experiments. N.H-S. and N.A. performed the cryo-EM analysis. T.S. and T.M. provided the technical support and conceptual advice. H.U. and H.N. conceived and supervised the study and wrote the manuscript. All the authors discussed the results and commented on the manuscript.

## Competing interests

The authors declare no competing interests.
