## [Transparent Peer Review file · Nature Communications]

Engineering of ATP synthase for enhancement of proton-to-ATP ratio

Corresponding Author: Dr Hiroshi Ueno

Version 0:

Reviewer comments:

Reviewer #1

(Remarks to the Author)

Hiroshi Ueno et al. conducted an engineering ATP synthase from *PS3* with the potentially up to three stalks, thus enhancing the proton-to-ATP ratio. The experimental design is innovative and comprehensive, especially in the functional analysis of proton-to-ATP ratio and the cryo-EM structure determinations. The manuscript is clearly written and the discussion is detailed. However, there are some issues to be further answered as follows:

1. Since the authors concluded that "ATP hydrolysis activity and proton pump activity were slower than those of the wild type", more discussions should be added on why this engineered protein complex could continue the synthase of ATP under low proton-to-ATP ratio.
2. As we know, the proton gradient across the membrane always exists. The authors used the FoF1-reconstituted proteoliposomes (PLs) for the functional determinations. Will it be different from the natural environment? It is curious to me what is the regulation mechanism of the critical value for ATP hydrolysis or synthase.
3. In the data processing, the authors first generated one NU-refinement map using all the selected particles. Have the authors tried to directly use the 1,584,329 particles to do the whole 3D classifications? The results could be compared and maybe several new classifications could be identified.
4. For Fig. 5, I recommend the authors could display the composite maps (Phenix-combine focused maps). This will be clearer.
5. I strongly recommend that all the softwares used in this article should label the specific version numbers, including cryoSPARC, ChimeraX, COOT, PHENIX, etc.

Reviewer #2

(Remarks to the Author)

Ueno and colleagues engineer F1Fo-ATP synthase from *Bacillus PS3* to introduce up to two additional peripheral stators, including the proton channel harboring 'a' subunits. To do so, the authors delete the N-terminal domain of the delta subunit and fuse its C-terminal domain to the alpha subunit N-terminus. SDS-PAGE of the purified preparation indicates that the mutant enzyme contains approximately twice the amount of 'b' and 'a' subunits compared to wild type. In the wild type enzyme, three catalytic nucleotide binding sites are coupled to a single proton channel, resulting in a H⁺ to ATP ratio of 3.3 (for the c10 ring of PS3). Functional characterization of the mutant complex reveals that the mutant enzyme can synthesize ATP at a significantly lower proton motive force (pmf; ~68 mV) compared to wild type (~133 mV), indicating functional incorporation of additional proton channel(s) in the mutant enzyme. By varying the mass action ratios of ATP, ADP and inorganic phosphate (Pi) and determining the equilibrium pmf values, they were able to obtain a H⁺ to ATP ratio of ~5.8 for the mutant enzyme, close to twice the value compared to wild type (~3). Overall, the data show that the mutant complex incorporated on average one additional peripheral stator including a coupled proton channel. To verify the functional conclusions, the authors determine cryoEM structures of the mutant preparation and they find complexes containing up to three peripheral stators/a-subunits, though the classes with two or three stators are not as populated as expected from the SDS-PAGE analysis. The authors speculate that this discrepancy is due to the instability of the complexes upon freezing for

cryoEM.

Overall, this is an interesting study that provides novel and important insight into the mechanism of proton transport-coupled ATP synthesis. The authors apply a clever protein engineering design to generate an enzyme that can synthesize ATP at a much lower pmf. The experimentation is of high quality and conclusions are well supported by the data. It will be interesting to see whether the organisms identified by the author based on genetic analysis have already established what the authors accomplished using protein engineering. The following lists a few suggestions which the authors may consider for improving the manuscript:

- (1) Line 69ff - The authors state that known c-ring stoichiometry ranges from 8 to 15, giving H⁺/ATP ratios of between 2.7 and 5. The authors may consider including Schulz et al., who recently reported a c-ring stoichiometry of 17 for the H⁺-coupled ATP synthase from *Burkholderia pseudomallei* (Schulz et al. EMBO Rep 18:526 (2017)), with a theoretical H⁺/ATP ratio of 5.7.
- (2) Figure 2: the ribbon representation makes it difficult to distinguish the different subunits - the figure could benefit from a similar rendering representation as used in Figure 1.
- (3) Line 245 "...two or three of which were not found in the wild type". Please clarify as wild type contains only one peripheral stator.
- (4) Line 258 - what are "cryo-EM cells"?
- (5) Line 340ff - The discussion about "four peripheral stalks" is confusing and needs clarification.
- (6) Line 355 - is the "well" in "well consistent" necessary?
- (7) Line 381 - "...suggesting that interference occurs among the a-subunits". This is unclear - please clarify.
- (8) Statistics: How many mutant preparations were analyzed? Are the H⁺-pumping and ATPase measurement repeats listed in Supplementary Table 1 ("n=6", "n=8") technical or biological repeats or a combination thereof? Similar question for the data shown in Figures 3 and 4.
- (9) The manuscript would benefit from editing for proper English syntax and grammar. For example, line 58 "couples" should be "couple"; line 75, "FoF1s ... and yeast mitochondria, which also..."; line 91 "...number of c-subunits ... number of a-subunits"; line 121 "...while the two others interact...", and many more.

Reviewer #3

(Remarks to the Author)
Ueno et al

The paper describes a remarkable, even extraordinary, piece of protein engineering where the ATP synthase from *Bacillus PS3* has been modified such that instead of it having the usual naturally occurring single peripheral stalk and attached static a-subunit with its two proton half channels in the interface between the a-subunit and the rotating c-ring, it has up to three such features. The consequences are profound since now the enzyme has a proton: ATP ratio of 5.8 instead of the natural value of $10/3 = 3.3$. In consequence, the enzyme can now operate at a much lower proton motive force than the natural bacterial enzyme. The data presented are convincing and the experiments have been carried carefully. The paper is written with great clarity.

The only difficulty I have is that in reality this extraordinary experiment, whilst exploiting and extending the range of possible structural and mechanistic features of the ATP synthase, actually tells us little that is new about how the enzyme works. On the other hand, it does help to solidify and confirm current knowledge by demonstrating that it provides a sound basis for the success of this astounding experiment, which may have practical exploitation. For this reason, I recommend acceptance

Minor points. Three important references have been omitted and should be included. They are:

1. Ferguson, S.F. ATP synthase: from sequence to ring size to P/O ratio. PNAS. 107 (39) 16755-16756
2. Stock et al. Molecular architecture of the rotary motor in ATP synthase Science (1999) 286, 1700-1705.
3. Watt et al. the bioenergetic cost of making an ATP molecule. PNAS (2010), 107, 16823-16827.

With the inclusion of these references, Reference 12 becomes redundant and it should be deleted.

Version 1:

Reviewer comments:

Reviewer #1

(Remarks to the Author)

The authors have carefully revised the manuscript and addressed all of the points raised by myself satisfactorily. As far as I can judge, also the points of reviewer 2 and reviewer 3 were addressed well.

Reviewer #2

(Remarks to the Author)

This is a revised manuscript. The authors' rebuttal and revisions to the manuscript, including information about statistics, inclusion of additional references, and improved figure presentation and writing have addressed the reviewer's critiques satisfactorily. The revisions have improved the manuscript significantly and the paper can now be recommended for publication in Nature Communications.

Reviewer #3

(Remarks to the Author)

Response to Reviewers' Comments

We sincerely appreciate valuable comments by the reviewers. According to the reviewer's comments, we have revised the manuscript. Changes are highlighted with **red color** in the highlighted version of the manuscript. With the help of the reviewers, we believe that the revised manuscript has been significantly improved and is suitable for publication in *Nature Communications*.

In addition to the revisions made in response to the reviewers' comments, we have also revised Figure 3b. We noticed that the band intensity of the *a*-subunit in both the wild-type and the $\delta_{\Delta N}$ - α fused F_oF₁ had been overestimated in the previous version of Figure 3b. Therefore, we recalculated the intensities and have revised Figure 3b accordingly. Importantly, even after recalculating, the stoichiometry of the *a*-subunit and *b*-subunit in the $\delta_{\Delta N}$ - α fused F_oF₁ remains approximately twice that of the wild-type. We sincerely appreciate the opportunity to refine our data presentation and improve the clarity of the manuscript.

Reviewers' comments are shown in *Italic*.

Reviewer #1

Comments:

Hiroshi Ueno et al. conducted an engineering ATP synthase from PS3 with the potentially up to three stalks, thus enhancing the proton-to-ATP ratio. The experimental design is innovative and comprehensive, especially in the functional analysis of proton-to-ATP ratio and the cryo-EM structure determinations.

Response:

We appreciate the positive evaluation by reviewer #1.

The manuscript is clearly written and the discussion is detailed. However, there are some issues to be further answered as follows:

1. Since the authors concluded that "ATP hydrolysis activity and proton pump activity were slower than those of the wild type", more discussions should be added on why this engineered protein complex could continue the synthase of ATP under low proton-to-ATP ratio.

Response:

We are not entirely certain how to interpret the comment, because the engineered F_oF₁ has a consistently higher H⁺/ATP ratio than the wild type. However, if we assume that "the synthase of ATP" refers to ATP synthesis, and that "under low proton-to-ATP ratio" refers to conditions of low *pmf*, our answer is as follows:

The key point is that the reaction rate (kinetics) and the equilibrium *pmf*, (thermodynamics) can be addressed independently. Specifically, the equilibrium *pmf*, is the point at which the rates of ATP synthesis and ATP hydrolysis are balanced; this is

fundamentally a thermodynamic parameter. It differs from the enzyme's kinetic characteristics, such as whether the engineered or the wild-type ATP synthase exhibits a higher or lower reaction rate. Thus, while the engineered ATP synthase may have a slower hydrolysis or synthesis rate, it can still drive ATP synthesis as long as a sufficient *pmf* is provided.

In our revised manuscript, we have included an additional paragraph for this point in the *Discussion* part (Lines 378-398).

2. As we know, the proton gradient across the membrane always exists. The authors used the FoF1-reconstituted proteoliposomes (PLs) for the functional determinations. Will it be different from the natural environment?

Response:

Proteoliposome system differs from the native cellular environment in that it provides a simplified and well-controlled platform for studying F_oF₁-ATP synthase. In preteoliposomes, purified F_oF₁-ATP synthase is reconstituted into liposomes. This setup allows precise control of the *pmf* on the liposome using acid-base transitions and the K⁺-valinomycin diffusion potential methods, while also enabling the control of the reaction quotient by adjusting ATP, ADP, and Pi concentrations. Hence, this system enables us to determine various fundamental properties of F_oF₁, such as the equilibrium *pmf*.

By contrast, in the native cellular environment, F_oF₁-ATP synthase operates under more “noisy” conditions. Cell membranes contain a wide variety of transporter molecules: some (including electron transport chains, ETCs) generate *pmf* by actively pumping protons, while others consume *pmf* through active or passive transport of substrates. Hence, the *pmf* in biological membranes is likely dynamic and constantly fluctuating, although intracellular pH is strongly buffered by numerous ionizable molecules. In addition, the reaction quotient, determined by intracellular concentrations of ATP, ADP, and Pi, is also subject to continuous change as these molecules are consumed or produced by various intracellular reactions. Therefore, F_oF₁-ATP synthase in cells is highly likely to experience more variable and dynamic conditions than in the reconstituted liposome system. Nonetheless, we wish to emphasize that the fundamental properties of the enzyme are, in principle, the same regardless of the environment, and should be measured under well-defined conditions such as those in proteoliposome experiments.

It is curious to me what is the regulation mechanism of the critical value for ATP hydrolysis or synthase.

Response:

Regarding the regulation mechanism of this enzyme, we note that it lies beyond the scope of the present work, although it is indeed an important question. Here, we reiterate that

the direction of the reaction—whether the enzyme catalyzes ATP hydrolysis or ATP synthesis—is primarily determined by thermodynamics: the balance between the proton motive force and the free energy of ATP hydrolysis, as shown in Equation (2) in the main text. If regulation systems alone were to determine the reaction direction, they would violate fundamental thermodynamic principles.

3. In the data processing, the authors first generated one NU-refinement map using all the selected particles. Have the authors tried to directly use the 1,584,329 particles to do the whole 3D classifications? The results could be compared and maybe several new classifications could be identified.

Response:

Thank you for your insightful suggestion regarding the 3D classification of all 1,584,329 particles without the initial NU-refinement. In response to your comment, we performed a direct 3D classification using all selected particles to explore whether additional structural classes could be identified. However, this approach did not yield distinct classes corresponding to each stalk structure. The primary challenge is thought to be that the classification process was predominantly driven by the high-resolution features of the F₁ portion rather than the lower-resolution stalk regions. Due to the greater structural flexibility and lower occupancy of stalk regions in the dataset, the classification algorithm tended to focus on the more stable and well-resolved regions of the complex. Consequently, the resulting 3D classes primarily reflected differences in the F₁ catalytic core rather than distinct stalk arrangements. In this study, we performed focused 3D classification using masks specifically targeting the peripheral stalk regions. This approach allowed us to resolve the different stalk conformations more effectively and identify distinct subpopulations with one, two, or three peripheral stalks.

4. For Fig. 5, I recommend the authors could display the composite maps (Phenix-combine focused maps). This will be clearer.

Response:

Thank you for your valuable suggestion. In response to this comment, we have added a composite maps to Fig. 5 to provide a clearer representation.

5. I strongly recommend that all the softwares used in this article should label the specific version numbers, including cryoSPARC, ChimeraX, COOT, PHENIX, etc.

Response:

Thank you for pointing, we labeled the specific version numbers of the softwares.

Reviewer #2

Comments:

Ueno and colleagues engineer F1Fo-ATP synthase from Bacillus PS3 to introduce up to two additional peripheral stators, including the proton channel harboring 'a' subunits. To do so, the authors delete the N-terminal domain of the delta subunit and fuse its C-terminal domain to the alpha subunit N-terminus. SDS-PAGE of the purified preparation indicates that the mutant enzyme contains approximately twice the amount of 'b' and 'a' subunits compared to wild type. In the wild type enzyme, three catalytic nucleotide binding sites are coupled to a single proton channel, resulting in a H⁺ to ATP ratio of 3.3 (for the c₁₀ ring of PS3). Functional characterization of the mutant complex reveals that the mutant enzyme can synthesize ATP at a significantly lower proton motive force (pmf; ~68 mV) compared to wild type (~133 mV), indicating functional incorporation of additional proton channel(s) in the mutant enzyme. By varying the mass action ratios of ATP, ADP and inorganic phosphate (Pi) and determining the equilibrium pmf values, they were able to obtain a H⁺ to ATP ratio of ~5.8 for the mutant enzyme, close to twice the value compared to wild type (~3). Overall, the data show that the mutant complex incorporated on average one additional peripheral stator including a coupled proton channel. To verify the functional conclusions, the authors determine cryoEM structures of the mutant preparation and they find complexes containing up to three peripheral stators/a-subunits, though the classes with two or three stators are not as populated as expected from the SDS-PAGE analysis. The authors speculate that this discrepancy is due to the instability of the complexes upon freezing for cryoEM.

Overall, this is an interesting study that provides novel and important insight into the mechanism of proton transport-coupled ATP synthesis. The authors apply a clever protein engineering design to generate an enzyme that can synthesize ATP at a much lower pmf. The experimentation is of high quality and conclusions are well supported by the data. It will be interesting to see whether the organisms identified by the author based on genetic analysis have already established what the authors accomplished using protein engineering.

Response:

We sincerely appreciate your positive evaluation.

The following lists a few suggestions which the authors may consider for improving the manuscript:

(1) Line 69ff - The authors state that known c-ring stoichiometry ranges from 8 to 15, giving H⁺/ATP ratios of between 2.7 and 5. The authors may consider including Schulz et al., who recently reported a c-ring stoichiometry of 17 for the H⁺-coupled ATP synthase from Burkholderia pseudomallei (Schulz et al. EMBO Rep 18:526 (2017)), with a theoretical H⁺/ATP ratio of 5.7.

Response:

We appreciate your comment. Indeed, before submitting our manuscript, we considered including information about the c₁₇-ring from *Burkholderia pseudomallei* (Schulz et al., EMBO Rep 18, 526 (2017)). However, this ATPase is not a canonical F-type ATPase; it

was recently classified as an N-type ATPase, which, while more similar to the F-type than to the V-type, exhibits distinct features. In light of your suggestion, we have now added a description of the c_{17} -ring and its theoretical H^+ /ATP ratio in the *Discussion* part of the revised manuscript (Lines 402-405).

(2) *Figure 2: the ribbon representation makes it difficult to distinguish the different subunits - the figure could benefit from a similar rendering representation as used in Figure 1.*

Response:

Thank you for the comment. We have revised the figure 2.

(3) *Line 245 "...two or three of which were not found in the wild type". Please clarify as wild type contains only one peripheral stator.*

Response:

Thank you for the comment. To clarify this point, we have revised the manuscript (Line 248).

(4) *Line 258 - what are "cryo-EM cells"?*

Response:

Thank you for the comment. We have revised the manuscript for clarity (Line 262).

(5) *Line 340ff - The discussion about "four peripheral stalks" is confusing and needs clarification.*

Response:

Thank you for the comment. To clarify this point, we have revised the manuscript (Lines 345-350).

(6) *Line 355 - is the "well" in "well consistent" necessary?*

Response:

Thank you for pointing this out. Along to the comment, we revised this phrase (Line 361).

(7) *Line 381 - "...suggesting that interference occurs among the a-subunits". This is unclear - please clarify.*

Response:

We have added the description to clarify this point (Lines 378-384).

(8) *Statistics: How many mutant preparations were analyzed? Are the H^+ -pumping and ATPase measurement repeats listed in Supplementary Table 1 ("n=6", "n=8") technical or biological repeats or a combination thereof? Similar question for the data shown in Figures 3 and 4.*

Response:

We have added these information in the figure legends.

(9) *The manuscript would benefit from editing for proper English syntax and grammar. For example, line 58 “couples” should be “couple”; line 75, “FoF1s ... and yeast mitochondria, which also...”; line 91 “...number of c-subunits ... number of a-subunits”; line 121 “...while the two others interact...”, and many more.*

Response:

Thank you for your valuable feedback regarding the English syntax and grammar in our manuscript. We have carefully revised the manuscript to address these issues.

Reviewer #3

The paper describes a remarkable, even extraordinary, piece of protein engineering where the ATP synthase from Bacillus PS3 has been modified such that instead of it having the usual naturally occurring single peripheral stalk and attached static a-subunit with its two proton half channels in the interface between the a-subunit and the rotating c-ring, it has up to three such features. The consequences are profound since now the enzyme has a proton: ATP ratio of 5.8 instead of the natural value of $10/3 = 3.3$. In consequence, the enzyme can now operate at a much lower proton motive force than the natural bacterial enzyme. The data presented are convincing and the experiments have been carried carefully. The paper is written with great clarity.

The only difficulty I have is that in reality this extraordinary experiment, whilst exploiting and extending the range of possible structural and mechanistic features of the ATP synthase, actually tells us little that is new about how the enzyme works. On the other hand, it does help to solidify and confirm current knowledge by demonstrating that it provides a sound basis for the success of this astounding experiment, which may have practical exploitation. For this reason, I recommend acceptance

Response:

We sincerely appreciate your positive evaluation.

Minor points. Three important references have been omitted and should be included. They are:

- 1. Ferguson, S.F. ATP synthase: from sequence to ring size to P/O ratio. PNAS. 107 (39) 16755-16756*
- 2. Stock et al. Molecular architecture of the rotary motor in ATP synthase Science (1999) 286, 1700-1705.*
- 3. Watt et al. the bioenergetic cost of making an ATP molecule. PNAS (2010), 107, 16823-16827.*

With the inclusion of these references, Reference 12 becomes redundant and it should be deleted.

Response:

According to the comments, we revised the references.